# Parallelization of a 3-Dimensional Hydrodynamics Model Using a Hybrid Method with MPI and OpenMP

**Jung Min Ahn** [1] , **Hongtae Kim** [1], **Jae Gab Cho** [2], **Taegu Kang** [1], **Yong-seok Kim** [3] **and Jungwook Kim** [1,*]

1   Water Quality Assessment Research Division, Water Environment Research Department,
    National Institute of Environmental Research, Incheon 22212, Korea; ahnjm80@gmail.com (J.M.A.);
    htkim8@korea.kr (H.K.); taegu98@korea.kr (T.K.)
2   Research and Development Institute, GeoSystem Research Corporation, Gunpo 15807, Korea;
    jgcho@GeoSR.com
3   Water Environment Research Department, National Institute of Environmental Research,
    Incheon 22689, Korea; nierkys@korea.kr
*   Correspondence: rlawjddnr1023@gmail.com; Tel.: +82-32-560-7477

**Abstract:** Process-based numerical models developed to perform hydraulic/hydrologic/water quality analysis of watersheds and rivers have become highly sophisticated, with a corresponding increase in their computation time. However, for incidents such as water pollution, rapid analysis and decision-making are critical. This paper proposes an optimized parallelization scheme to reduce the computation time of the Environmental Fluid Dynamics Code-National Institute of Environmental Research (EFDC-NIER) model, which has been continuously developed for water pollution or algal bloom prediction in rivers. An existing source code and a parallel computational code with open multi-processing (OpenMP) and a message passing interface (MPI) were optimized, and their computation times compared. Subsequently, the simulation results for the existing EFDC model and the model with the parallel computation code were compared. Furthermore, the optimal parallel combination for hybrid parallel computation was evaluated by comparing the simulation time based on the number of cores and threads. When code parallelization was applied, the performance improved by a factor of approximately five compared to the existing source code. Thus, if the parallel computational source code applied in this study is used, urgent decision-making will be easier for events such as water pollution incidents.

**Keywords:** EFDC-NIER; parallel calculation; optimization; OpenMP; MPI

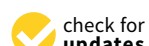

## 1. Introduction

In areas with monsoon climates, including South Korea, changes in hydraulic characteristics throughout the year are significant because weather conditions such as precipitation and air temperature change throughout the year [1,2]. The flow rate is strong during the flood season, but weak during the dry season. During the summer, when the temperature is high, stratification develops and sometimes causes various environmental problems [3]. In particular, the multi-functional weirs built as part of the Four Major River Restoration Project increase the residence time and water depth, which deepens the stratification phenomenon.

Changes in hydraulic characteristics also affect water quality [4]. Therefore, the vertical layers need to be classified to precisely simulate hydraulics and water quality for these conditions. Furthermore, cyanobacteria, which are generated in large quantities during the summer, often have different distribution characteristics along the lateral direction of a river, and the water quality distribution worsens downstream around basic environmental facilities or confluences of major pollutant sources. Vertically integrated 2D or x-z 2D models are limited in their ability to reproduce these 3D variation characteristics. Therefore, the application of a 3D model with an appropriate vertical/horizontal resolution is required,

but the computational requirements can increase significantly if a model is constructed using 3D high-resolution grids [5]. In particular, the computational requirements increase sharply if a long-term simulation is performed for an entire river, such as the Nakdong River. To overcome this problem, it is necessary to reduce the time required to conduct the calculations by adjusting the number of calculations or calculation intervals using certain techniques, such as parallel calculation, independent computations of a hydraulic-water quality model, or application of an implicit scheme.

In particular, research on parallelization is very important in the field of modeling. With advancements in computer performance, it is possible to simulate various phenomena on a large scale and over an extended period. However, the number of input data items required for the model has increased. Furthermore, the number of parameters of the model required for simulation has also increased. Most of the models currently in use are executed serially, resulting in the simulation taking a long time to complete [6]. The evolution of the model conversely slows down the process. To overcome this issue, the model must be parallelized. The large-scale hydrological model simulates the water resources, changes in water quality, and water circulation globally. Therefore, much data such as weather, climate, runoff, and topography are required for modeling. For this reason, previous studies have tried to reduce the simulation speed by applying a parallelization technique to a large-scale basin model. Neal et al. [7] applied three parallelization techniques based on OpenMP, message passing, and specialized accelerator cards to improve the simulation speed of a 2D flood inundation model that requires various input data. Rouholahnejad et al. [8] significantly reduced the time to calibrate parameters by parallelizing the Soil and Water Assessment Tool (SWAT) model. Liu et al. [9] improved the simulation speed by parallelizing the grid-based distributed hydrological model with OpenMP. Avesani et al. [6] developed HYPERstreamHS, which parallelizes a large-scale river basin model to efficiently consider various hydraulic structures. In particular, the parallel performance was better in the large-scale model than in the small-scale model [9,10].

Parallel calculation is an efficient calculation technique that uses multiple resources simultaneously through job allocation and data distribution via a communication method, whereby processes with separate local memories send or receive data for sharing. A parallel calculation should be divided into separate task fragments that can be solved simultaneously, and should be solved in less time using multiple calculation resources rather than a single one. For example, Barney [11] proposed a conceptual scheme for parallel computing performed by multiple processes by subdividing a problem. Parallel computing methods include message passing interfaces (MPIs) based on a distributed memory system [12], open multi-processing (OpenMP) based on shared memory [13], and a method based on manycore processors such as an Intel Xeon or a graphics processing unit (GPU). Each parallel computing method has advantages and disadvantages. OpenMP is simple to implement, but it has limited scalability, depending on the size of the shared memory system. In contrast, MPI is scalable for high-dimensional problems, but its computing efficiency tends to decrease as the number of CPU cores increases because of the increase in communication between the CPUs. Thus, a high-performance coding technique is required to solve this problem effectively. In the parallelization of manycore processors such as GPUs, the device dependency is high, and the performance is improved only for certain types of calculations. In recent years, studies have been actively conducted on hybrid techniques that combine the OpenMP and MPI methods [14–20]. Recently, there has been a study that applied a hybrid technique combining OpenMP and MPI to an urban flood model. It was confirmed that the parallelization ability was better than with OpenMP or MPI solely [21].

Among various process-based numerical models, the Environmental Fluid Dynamics Code (EFDC) was developed with the FORTRAN language by the Virginia Institute of Marine Science (VIMS) in the United States, and the Environmental Protection Agency (EPA) released a generalized vertical grid (GVC) version [22–25]. The numerical scheme employed in EFDC to solve the equations of motion uses a second-order accurate spatial

finite difference on a staggered or C grid, and the EFDC model's internal momentum equation solution, at the same time step as the external, is implicit with respect to vertical diffusion [25]. See reference [25] for more information about the model concept, etc.

Dynamic Solutions International (DSI) released the EFDC_DS (20100328 version), and has been continually updating the source code since then. The EFDC model is used worldwide, which demonstrates its performance and applicability.

Based on the EFDC_DS (20100328 version), the National Institute of Environmental Research (NIER) in South Korea has added more features, such as the operating function of hydraulic structures in the major rivers of South Korea; a simulation function of multi-algae species; a vertical migration mechanism for cyanobacteria; akinete generation and germination mechanisms; and a mechanism for the effect of salt and toxicity on freshwater and sea algae, wind stress, and benthic flux of inorganic nutrients according to changes in the oxidation/reduction conditions [26].

Ahn et al. [26,27] established a method for short-term prediction of algae by using an improved source code with an operating function of hydraulic river structures and of mechanisms for vertical migration of cyanobacteria and akinete creation and germination. They also proposed an optimal method for predicting algae by applying hyperspectral remote-sensing data to the EFDC-NIER model. The model, which the NIER has named EFDC-NIER, has improved the functions to suit the major rivers of South Korea, and it is used to support the policies for algae and water quality control of major rivers and lakes in South Korea. Because the number of calculations required for analysis has increased in various fields such as hydraulics, hydrology, water quality, and aquatic ecosystems, a parallel computational code needs to be applied to increase the utilization of policies through fast decision-making.

In this study, we applied a hybrid parallel computational code constructed using the OpenMP and MPI methods to the EFDC-NIER model, and compared the calculation times required for the existing version and the parallel computational code version. We also compared the simulation results for the existing EFDC model and the parallel computational code model to check if they corresponded. Furthermore, the optimal parallel combination was determined through a comparative evaluation of the simulation time based on the number of cores and threads.

## 2. Materials and Methods

### 2.1. Research Trend in Parallel Calculation of EFDC Model

The EFDC is a general-purpose modeling package used for the simulation of 3D flow, material transport, and biochemical processes in systems such as rivers, lakes, estuaries, reservoirs, wetlands, and coasts. The EFDC model is open-source software developed by John Hamrick at the Virginia Institute of Marine Science (VIMS, Gloucester point, VA, USA) [28]. The EFDC is one of the models recommended by the US EPA for the management of the total maximum daily load (TMDL), and the US EPA continues to support its development. The EFDC has been extensively tested and used by many researchers in several modeling studies.

DSI has developed a version of the code that streamlines the modeling process and connects it to the pre- and post-processing tools of DSI [29]. DSI has developed and commercialized the EFDC_DSI OpenMP code by applying the OpenMP library for parallel computing. Figure 1 shows the comparison between the simulation times when the EFDC_DSI OpenMP is applied [30]. The model using an octa-core processor reduced the execution time of all subroutines by approximately 75% compared to the model that used a single-core processor (the dotted line on the left). However, although the calculation time of the subroutine that simulated the water quality response mechanism decreased slightly when a dual-core processor was used, the calculation time was similar to that when using a single-core processor. As the number of processors increased (the dotted line on the right), the calculation time did not significantly decrease.

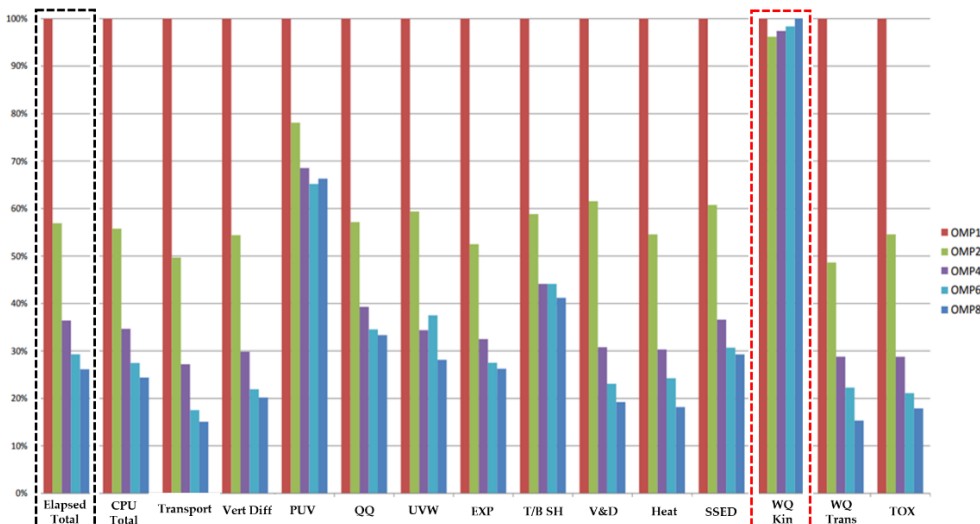

**Figure 1.** Comparison of simulation times when the EFDC_DSI OpenMP is applied (Vert Diff: Vertical Diffusion; PUV: Pressure, U, and V flux; QQ: Turbulent intensity; UVW: U, V, and W velocity; EXP: Explicit momentum equation term; T/B SH: Bottom friction; V&D: Viscosity and Diffusivity; Heat: Heat flux; SSED: Sediment transport; WQ Kin: WQ kinetic equation; WQ Trans: WQ transport; TOX: Toxic).

IBM used the MPI library to parallelize the GVC version of the EFDC model released by the EPA [31] and published it on GitHub [32]. For continuous management during a parallelization operation, the setup process for parallel execution was automated (1) to limit the changes in a large number of source files to avoid computational errors, (2) to ensure that the results of serial calculations and parallel calculations match, and (3) to ensure that the originally configured serial model runs properly in the parallel code. Figure 2 shows the parallel efficiency as a function of the number of processors using IBM's parallelized code. When six processors were used, the parallel efficiency was 50%, and when 25 processors were used, an efficiency of 40% was achieved. However, the rate of increase in the efficiency fell when more than six processors were used. The parallel efficiency for the water quality response calculation was unknown because it was calculated by parallelizing only the hydraulic module from the tidal current model of Galway Bay.

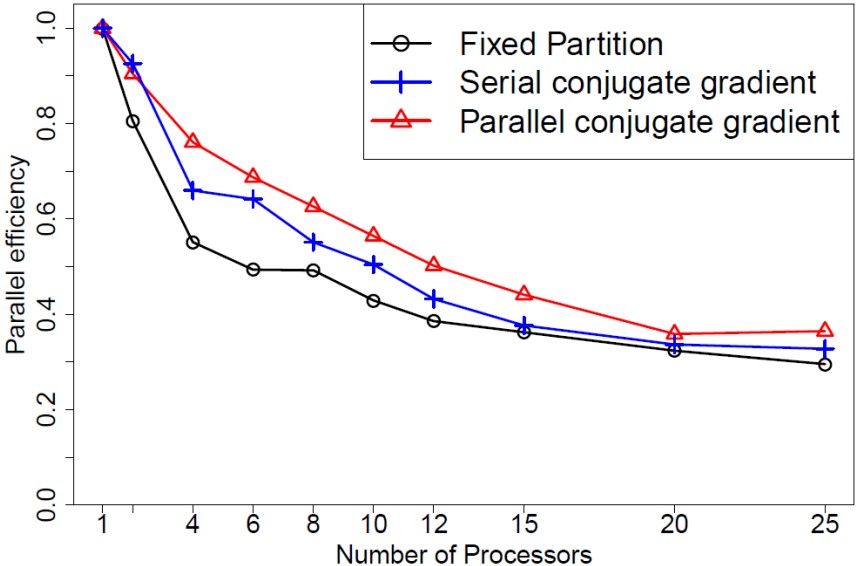

**Figure 2.** Parallel efficiency as a function of the number of processors using IBM's MPI-parallelized code.

*2.2. Development of Parallel Computational Code*

As described above, DSI performed parallelization using the OpenMP method, and IBM performed parallelization using the MPI method. IBM performed parallelization only on the subroutines related to the hydraulic calculations. Therefore, in this study, we aimed to apply the parallel code to all of the EFDC-NIER model based on a hybrid method of applying both OpenMP and MPI. Because both MPI and OpenMP have advantages and disadvantages, the hybrid method can utilize the advantages of both while minimizing their drawbacks. In hybrid parallel programming, an intensive calculation is performed at a single node using the OpenMP method, and a large number of calculations are performed by executing communications between different nodes over the network based on the MPI method. In 2019, a research team at the University of Bialystok in Poland conducted a study on the K-means parallelization algorithm. Statistics such as the vector sum and count were calculated using OpenMP at each MPI node, and after gathering them at the master node, the remaining statistical calculations were completed and delivered to each processor to continue the next calculation. Similarly, four algorithms were parallelized, and the results of the computational experiments showed that all these algorithms were superior to the conventional Lloyd algorithm in terms of computing time [33].

MPI is a library specification containing standardized subroutines and functions for handling communication [34], whereas OpenMP is a shared memory model and an add-on to a compiler. One of the advantages of OpenMP is that there is no communication between nodes, but the disadvantage is that the user's desired calculation time cannot be secured when large calculations are performed because a single node cannot be extended infinitely. In contrast, in the case of MPI, calculation nodes can be added to secure the calculation speed desired by the user.

Parallel code development was carried out in two stages: Sequential code optimization and parallel code creation. Sequential code optimization refers to the process of maximizing the performance of the sequential code before parallelizing the code (Table A1). The optimization was performed by finding unnecessary statements through measuring the precise computation time for each computational statement in the subroutines defined earlier by hotspot analysis. A DEBUG variable was added to the CALUVW subroutine to compose a log fine (CFL.OUT) when DEBUG was required, and the variable I/O was reduced in EEXPOUT by reducing the loop depth to 3 (NSP, K, L) when writing the result.

Parallel codes were developed using a hybrid parallel programming method that included both OpenMP and MPI. They were created by adding statements used by both parallel libraries, and parallelization was performed for approximately 40 source codes. For parallel code development, we chose the hybrid parallel programming approach and developed a technique for dividing the computational area according to the processor used. The MPI parallel code was composed by partitioning the LA variable (the number of calculation grids) corresponding to the 2D index in a Do loop, and the OpenMP parallel code was composed using a thread fork-join approach with the partitioned indices. To aid in the understanding of how the hybrid parallel code is created, Table A2 shows an example of a parallel code. First, we designed it such that the statements repeated from 2 to LA in the Do loop of the sequential code would be repeated from LMPI2 to LMPILA and partitioned according to the MPI rank in the parallel code. We then parallelized each Do loop using !$OMP PARALLEL DO, OpenMP Directive. The RS8Q variable is configured to calculate the sum of all indices by simultaneously using OpenMP's REDUCTION and MPI's ALLREDUCE.

We developed MPI functions for area partitioning and communication according to the number of MPI nodes, and Table 1 presents the specific roles of each function. Among them, the BROADCAST_BOUNDARY function was developed to communicate the updated boundary values between the MPI nodes to and from adjacent nodes. The COLLECT_IN_ZERO function transmits variables from all nodes to the master node in order to output the results and analyze the consistency at the master node. For the data type used in the MPI communication, a user-defined data type (MPI_TYPE_VECTOR)

was used to reduce the frequency and amount of communication. For the topology, a single topology without partitioning was used. MPI-only functions that handled the communication between nodes were inserted to send and receive the boundary values for a minimal number of times; otherwise, a communication load would occur.

**Table 1.** MPI-only functions of the EFDC-NIER parallel code.

| Function | Description |
|---|---|
| MPI_INITIALIZE | Initializes MPI and sets up MPI variables (number of nodes, rank of each node) |
| MPI_DECOMPOSITION | Partitions the LA index |
| BROADCAST_BOUNDARY | Performs communication of boundary values between MPI nodes (1D variable) |
| BROADCAST_BOUNDARY_ARRAY | Performs communication of boundary values between MPI nodes (2D and 3D) |
| COLLECT_IN_ZERO | Performs collective communication to send the variables to the master node (1D) |
| COLLECT_IN_ZERO_ARRAY | Performs collective communication to send the variables to the master node (2D and 3D) |
| MPI_TIC/MPI_TOC | Timestamp for hotspot analysis (MPI Walltime) |

*2.3. Parallel Calculation Test Model Sets*

To evaluate the parallelization performance and consistency of the EFDC-NIER according to the number of grids, we used the model sets presented in Tables 2 and 3. Measuring the consistency of the parallel code involves evaluating whether the results for the serial and parallel calculations match.

**Table 2.** Method for evaluating the parallelization performance.

| Case | | Description | Comparison Method |
|---|---|---|---|
| Case 1 | | Before optimization and parallel code application | |
| Case 2 | 1 | Sequential code optimization | |
| | 2 | Case 2-1 + Changed method of writing the result file | |
| Case 3 | | Case 2-2 + OpenMP (6 threads) | Consistency of calculation times and results for simulations from 1 July 2015 to 9 July 2015 |
| Case 4 | | Case 2-2 + MPI (6 nodes) | |
| Case 5 | 1 | Case 2-2 + Hybrid (1 thread + 5 nodes) | |
| | 2 | Case 2-2 + Hybrid (2 threads + 4 nodes) | |
| | 3 | Case 2-2 + Hybrid (3 threads + 3 nodes) | |
| | 4 | Case 2-2 + Hybrid (4 threads + 2 nodes) | |
| | 5 | Case 2-2 + Hybrid (5 threads + 1 node) | |

**Table 3.** Model and computer specifications used for the evaluation of the parallel calculation.

| Category | | Description |
|---|---|---|
| Model | Number of grids | Horizontal: 6998 units, vertical: 11 layers |
| Windows PC | CPU | Inter Core i7-8700 3.20 GHz, 6 cores |
| | Memory | 32 GB |
| | OS | Microsoft Windows 10 Pro 64-bit |
| | Compiler | Intel parallel studio XE 2020, 19.1.2.254 20200623 |

## 3. Results and Discussion

### 3.1. Source Code Analysis for the EFDC-NIER Model

In models such as EFDC-NIER, which rely on data-intensive calculations, it is important to identify the time-consuming calculation codes in the program before creating the parallelization code. Such a time-consuming function or section is called a hotspot, and it is crucial to clearly identify hotspots to perform optimization and parallelization for overall performance improvement.

In this study, we identified 20 subroutines and calculation statements as hotspots among approximately 250 subroutines executed in the source code of EFDC-NIER. Table 4 presents the names of the subroutines identified as hotspots, their execution times, and the proportions of the total execution time for each subroutine. The HDMT2T subroutine is the main execution component of the numerical operations, and for the 20 hotspots, the proportions are calculated by comparing their execution time to the execution time of HDMT2T. Twelve subroutines accounted for more than 1% of the total execution time: Water quality component simulation (WQ3D) > flow rate/direction component simulation (CALUVW) > concentration (water temperature, sediment) simulation (CALCONC) > subroutine description (CALQQ2T) > subroutine description (CALEXP2T). The proportions of these subroutines were high because they simulated 3D variables.

**Table 4.** Hotspot analysis results for the EFDC-NIER model.

| Function Name | Description | Proportion of Execution Time (%) |
|---|---|---|
| HDMT2T | Main code of numerical simulation | 100.00 |
| WQ3D | Water quality component simulation | 38.03 |
| CALUVW | Flow rate/direction component simulation | 18.45 |
| CALCONC | Concentration (water temperature, sediment) simulation | 10.65 |
| CALQQ2T | Turbulence intensity simulation | 7.33 |
| CALEXP2T | Explicit momentum equation calculation | 6.17 |
| CALHDMF | Simulation of horizontal viscosity and spreading momentum | 4.66 |
| CALPUV2C | Simulation of surface P, UHDYE, and VHDXE | 4.12 |
| EEXPOUT | Writing a result file | 3.15 |
| CALAVB | Simulation of vertical viscosity and dispersion | 3.12 |
| ADVANCE | Updating the next timestep value | 0.98 |
| CALBUOY | Buoyancy simulation | 0.98 |
| CALTBXY | Bottom friction factor calculation | 0.60 |
| QQSQR | Updating the turbulence intensity value | 0.56 |
| CALCSER | Updating the boundary data | 0.35 |
| CALTSXY | Updating the surface wind stress | 0.28 |
| SEDIMENT | Sediment simulation | 0.25 |
| NLEVEL | Distribution of variables for each timestep | 0.11 |
| SALPTH | Writing the salinity result | 0.09 |
| DUMP | Recording the model variable dump | 0.06 |

### 3.2. Parallel Performance Evaluation

For the parallel performance evaluation, we compared the execution times of the model for the cases presented in Table 2 and examined the execution times and performance improvement rates of the major subroutines for each parallel combination.

When the simulation times for each case were compared, the source code optimization and parallel code application resulted in a performance improvement by a factor of approximately five compared to Case 1 (Table 5). When the optimization was performed for EEXPOUT (shown in Table A1), the time decreased by approximately 1.5 h, indicating that a significant amount of time was spent in the writing method. A comparison of OpenMP and MPI showed that the MPI method required 0.05 h less in the simulation time. However, in the case of personal computers, it would be sufficient to use OpenMP because the core resource is limited.

**Table 5.** Parallelization performance evaluation method results.

| Case | | Simulation Time (h) |
|---|---|---|
| Case 1 | | 3.22 |
| Case 2 | 1 | 2.68 |
| | 2 | 1.69 |
| Case 3 | | 0.74 |
| Case 4 | | 0.69 |
| Case 5 | 1 | 0.66 |
| | 2 | 0.73 |
| | 3 | 0.68 |
| | 4 | 0.65 |
| | 5 | 0.79 |

For the combination of OpenMP and MPI, we compared the simulation times by increasing the number of OpenMP threads from one to six and decreasing the number of MPI nodes from five to one in each case. When four threads of OpenMP and two nodes of MPI were applied, the fastest simulation was executed in 0.65 h. It is difficult to determine whether a certain combination of OpenMP thread count and MPI node count is sufficient, and an appropriate combination that accounts for the specifications of the user's computer should be determined.

We conducted this study using a Windows PC, which offers versatility. OpenMP can use only the cores corresponding to one computer, whereas MPI can utilize all cores of multiple computers. Therefore, the performance improvement will increase further if the software is applied to a Linux-based supercomputer. Appendix B describes the evaluation results for the parallel computing performance in a Linux-based cluster, and Appendix C describes the evaluation results for the parallel computing performance of the OpenMP and MPI methods.

### 3.3. Consistency Evaluation

In the process of parallelizing the sequential code, two categories can be considered for the factors that violate the consistency of the parallelization code. A complete understanding of the code is required before developing the parallelization code. If parallel programming directives are added without sufficient understanding of the algorithm, the variables that need to be communicated may not be communicated, resulting in improper synchronization. Additionally, subroutines that are calculated at a certain period may not be recognized, resulting in the omission of their parallelization. These cases imply that the parallel code has not been written properly, and a sufficient understanding of the code should be gained before proceeding.

In the second case, because a difference occurs in the order of calculations in the process of parallelizing the sequential code, a rounding error may occur owing to the limitations of floating-point arithmetic operations. Rounding errors occurred in the CONGRAD subroutine in the development process of the parallelization code for EFDC-NIER. To resolve this problem, we changed the types of variables (RPCG, PAPCG, RPCGN, ALPHA, and BETA) used for the calculations from the REAL 4-byte type to the REAL 8-byte type. The consistency of the parallel code means that the parallel execution results in the simulation are the same as the results of the sequential model for the simulation domains and the simulation options that the user can use. We considered the errors of the floating-point arithmetic operations when determining whether the simulation produced the same results. The consistency was evaluated by comparing the sum of the absolute values of all matrix values of the variables simulated up to a certain prediction period based on the variables that were written to the results file. As shown in Tables 6 and 7, the consistency evaluation results confirm that the results exactly match the sequential execution of the parallel code, execution of the OpenMP, and execution of the MPI.

**Table 6.** Consistency analysis results for each evaluation subject.

| Variable | Case 1 | Case 3 | Case 4 |
|---|---|---|---|
| TSX | $6.173959 \times 10^{-4}$ | $6.173959 \times 10^{-4}$ | $6.173959 \times 10^{-4}$ |
| TSY | $6.763723 \times 10^{-4}$ | $6.763723 \times 10^{-4}$ | $6.763723 \times 10^{-4}$ |
| TBX | $2.715987 \times 10^{-2}$ | $2.715987 \times 10^{-2}$ | $2.715987 \times 10^{-2}$ |
| TBY | $3.796839 \times 10^{-1}$ | $3.796839 \times 10^{-1}$ | $3.796839 \times 10^{-1}$ |
| AV | 6.224335 | 6.224335 | 6.224335 |
| AB | 7.667015 | 7.667015 | 7.667015 |
| AQ | 2.551961 | 2.551961 | 2.551961 |
| HP | $3.046219 \times 10^{4}$ | $3.046219 \times 10^{4}$ | $3.046219 \times 10^{4}$ |
| HU | $3.080725 \times 10^{4}$ | $3.080725 \times 10^{4}$ | $3.080725 \times 10^{4}$ |
| HV | $3.018550 \times 10^{4}$ | $3.018550 \times 10^{4}$ | $3.018550 \times 10^{4}$ |
| P | $1.726112 \times 10^{6}$ | $1.726112 \times 10^{6}$ | $1.726112 \times 10^{6}$ |
| U | $1.927491 \times 10^{2}$ | $1.927491 \times 10^{2}$ | $1.927491 \times 10^{2}$ |
| V | $1.632354 \times 10^{3}$ | $1.632354 \times 10^{3}$ | $1.632354 \times 10^{3}$ |
| W | $8.197821 \times 10^{-1}$ | $8.197821 \times 10^{-1}$ | $8.197821 \times 10^{-1}$ |
| TEM | $1.677452 \times 10^{5}$ | $1.677452 \times 10^{5}$ | $1.677452 \times 10^{5}$ |
| SEDT | $4.259870 \times 10^{5}$ | $4.259870 \times 10^{5}$ | $4.259870 \times 10^{5}$ |
| QQ | $1.507725 \times 10$ | $1.507725 \times 10$ | $1.507725 \times 10$ |
| QQL | 1.070996 | 1.070996 | 1.070996 |
| WQV | $2.552020 \times 10^{5}$ | $2.552020 \times 10^{5}$ | $2.552020 \times 10^{5}$ |
| WQVX | $5.815869 \times 10^{4}$ | $5.815869 \times 10^{4}$ | $5.815869 \times 10^{4}$ |

**Table 7.** Consistency results of major water quality factors.

| Date | Case 1 | | | Case 3 | | | Case 4 | | | Case 5-4 | | |
|------|--------|------|------|--------|------|------|--------|------|------|----------|------|------|
| | BOD | T-N | T-P | BOD | T-N | T-P | BOD | T-N | T-P | BOD | T-N | T-P |
| 01-07-2015 | 1.453 | 3.590 | 0.080 | 1.453 | 3.590 | 0.080 | 1.453 | 3.590 | 0.080 | 1.453 | 3.590 | 0.080 |
| 02-07-2015 | 1.481 | 3.609 | 0.079 | 1.481 | 3.609 | 0.079 | 1.481 | 3.609 | 0.079 | 1.481 | 3.609 | 0.079 |
| 03-07-2015 | 1.521 | 3.658 | 0.080 | 1.521 | 3.658 | 0.080 | 1.521 | 3.658 | 0.080 | 1.521 | 3.658 | 0.080 |
| 04-07-2015 | 1.547 | 3.700 | 0.081 | 1.547 | 3.700 | 0.081 | 1.547 | 3.700 | 0.081 | 1.547 | 3.700 | 0.081 |
| 05-07-2015 | 1.520 | 3.712 | 0.079 | 1.520 | 3.712 | 0.079 | 1.520 | 3.712 | 0.079 | 1.520 | 3.712 | 0.079 |
| 06-07-2015 | 1.596 | 3.672 | 0.081 | 1.596 | 3.672 | 0.081 | 1.596 | 3.672 | 0.081 | 1.596 | 3.672 | 0.081 |
| 07-07-2015 | 1.637 | 3.612 | 0.080 | 1.637 | 3.612 | 0.080 | 1.637 | 3.612 | 0.080 | 1.637 | 3.612 | 0.080 |
| 08-07-2015 | 1.563 | 3.568 | 0.081 | 1.563 | 3.568 | 0.081 | 1.563 | 3.568 | 0.081 | 1.563 | 3.568 | 0.081 |
| 09-07-2015 | 1.634 | 3.493 | 0.081 | 1.634 | 3.493 | 0.081 | 1.634 | 3.493 | 0.081 | 1.634 | 3.493 | 0.081 |

## 4. Conclusions

In this study, we optimized the source code of the EFDC-NIER model, which has been enhanced by the National Institute of Environmental Research (NIER) since 2010, and applied a parallel computational code. Then, we examined the consistency of the results between the existing EFDC model and the parallel computational code simulations and compared their calculation times. Furthermore, we determined an optimal parallel combination through a comparative evaluation of the simulation times between the number of cores and threads. The major findings of this study are as follows.

(1) The source code optimization and parallel code application resulted in a performance improvement by a factor of approximately five compared to the existing source code (Case 1). In the case of the existing EFDC, a large amount of time was consumed by a subroutine that wrote the results, and when this was improved, it took approximately half as much time for the calculation. As shown in Appendix B, the parallel calculation performance of the OpenMP and MPI methods applied in this study showed a similar level of performance as the results of the version developed and released by DSI. Therefore, the parallel calculation of the EFDC-NIER is better than or on par with that of the EFDC+ MPI, especially considering that its improvement values include the simulation results of the water quality factors.

(2) For a Windows PC, there is no difference in the reduction of the calculation speed between the OpenMP and MPI methods because the core and thread resources are limited. However, as shown in Appendix C, when using a Linux server, the simulation is performed more ideally when the MPI method is used compared to the OpenMP method. In the case of the hybrid method that uses both the OpenMP and MPI methods, the optimal computing combination should be applied according to the performance and computing resources of the computer on which the simulation will be performed.

(3) In the case of South Korea, algae prediction information for the water supply source sections of large rivers is sent to the water quality managers at eight-day intervals. When predicting the algae eight days into the future, all prediction work must be completed within 8 h. The fastest simulation's calculation time (0.65 h) is a very important factor; if the optimization and the parallel computational source code applied in this study are used, quick calculation will be facilitated when urgent decision-making is required for an event such as a water pollution incident.

**Author Contributions:** Conceptualization, J.M.A.; methodology, J.M.A., H.K. and J.K.; software, J.M.A.; validation, J.M.A. and J.K.; formal analysis, J.M.A. and J.K.; investigation, J.M.A., H.K., J.G.C. and T.K.; resources, T.K. and Y.-s.K.; data curation, J.M.A., Y.-s.K. and J.K.; writing—original draft preparation, J.M.A.; writing—review and editing, J.K.; visualization, H.K. and J.G.C.; supervision, J.K.; project administration, J.M.A. and J.K.; funding acquisition, T.K. and Y.-s.K. All authors have read and agreed to the published version of the manuscript.

**Funding:** This research was funded by a grant from the National Institute of Environmental Research (NIER) (grant number NIER-2020-01-01-012), which is funded by the Ministry of Environment (MOE) of the Republic of Korea.

**Institutional Review Board Statement:** Not applicable.

**Informed Consent Statement:** Not applicable.

**Data Availability Statement:** The data presented in this study are available on request from the corresponding author.

**Acknowledgments:** This study was supported by a grant (NIER-2020-01-01-012) from the National Institute of Environmental Research (NIER), which is funded by the Ministry of Environment (MOE) of the Republic of Korea.

**Conflicts of Interest:** The authors declare no conflict of interest.

## Appendix A

**Table A1.** Examples of source code optimization.

| | Variables | | Code Description |
|---|---|---|---|
| 1. | CALUVW | Improved | IF (ISCFL.GE.1.AND.DEBUG) THEN<br>IF (MYRANK.EQ.0) THEN<br>OPEN (1, FILE = 'CFL.OUT', STATUS = 'UNKNOWN', POSITION = 'APPEND')<br>ENDIF<br>IF (MYRANK.EQ.0) THEN<br>IF (ISCFL.EQ.1) WRITE (1,1212) DTCFL, N, ICFL, JCFL, KCFL<br>IF (ISCFL.GE.2.AND.IVAL.EQ.0) WRITE (1,1213) IDTCFL<br>ENDIF<br>ENDIF |
| 2. | EEXPOUT | Conventional | DO NSP = 1, NXSP<br>DO K = 1, KC; DO L = 2, LA<br>WQ = WQVX(L, K, NSP)<br>WRITE (95) WQ<br>ENDDO; ENDDO<br>ENDDO |
| | | Improved | DO NSP = 1, NXSP<br>WRITE (95) WQVX (:,:,NSP)<br>ENDDO |

**Table A2.** Example of hybrid parallel code application.

| Category | Description |
|---|---|
| Sequential code | DO L = 2,LA<br>RCG_R8(L) = CCC(L)*P(L) + CCS(L)*PSOUTH(L) + CCN(L)*PNORTH(L)<br>& + CCW(L)*P(L-1) + CCE(L)*P(L + 1)-FPTMP(L)<br>ENDDO<br>DO L = 2,LA<br>RS8Q = RS8Q + RCG_R8(L)*RCG_R8(L)<br>ENDDO |
| Parallel code | !$OMP PARALLEL DO<br>DO L = LMPI2,LMPILA<br>RCG_R8(L) = CCC(L)*P(L) + CCS(L)*PSOUTH(L) + CCN(L)*PNORTH(L)<br>& +CCW(L)*P(L-1) + CCE(L)*P(L + 1)-FPTMP(L)<br>ENDDO<br>!$OMP PARALLEL DO REDUCTION(+:RS8Q)<br>DO L = LMPI2,LMPILA<br>RS8Q = RS8Q + RCG_R8(L)*RCG_R8(L)<br>ENDDO<br>CALL MPI_ALLREDUCE(RS8Q,MPI_R8,1,MPI_DOUBLE,<br>& MPI_SUM,MPI_COMM_WORLD,IERR)<br>RS8Q = MPI_R8 |

**Appendix B**

DSI, the development and distribution organization of the EFDC model and the EFDC-Explorer, presented the results of the development and performance of the MPI-based EFDC+ in July 2020 [35]. The previous EFDC+ performed parallelization using OpenMP, and at the time, the calculation speed improvement was a factor of 2.5. This is similar to the performance improvement factor of the OpenMP performed in this study, which is approximately a factor of two to three. In the case of OpenMP, improvement in speed is impossible without the improvement of the CPU because it is affected by the performance of the computing resources. However, MPI is more favorable for speed calculation improvement than OpenMP because it is possible to use distributed computing resources, although the improvement varies depending on the constructed model. As a result of applying the source code optimization and the OpenMP and MPI methods in this study, the calculation speed improved by a factor of approximately five. In the case of OpenMP, the calculation speed improvement was similar for different numbers of calculation grids, but in the case of MPI, the calculation speed improved as the number of grids increased.

In addition, we compared the results of using similar processors to evaluate the MPI parallel code of EFDC+, developed by DSI. Table A3 presents an overview of the model and specifications of the hardware used. The number of horizontal grids is approximately 120,000 in EFDC-NIER and approximately 200,000 in EFDC+. However, the number of vertical layers was 11 in EFDC-NIER and four in EFDC+. DSI's EFDC+ provides the simulation results for only the hydraulic factors, and the EFDC-NIER produces the results by performing parallelization of the hydraulics and the water quality. Table A4 presents the performance improvement in the calculation speed for each MPI processor. Because numerical results are not provided in the report for EFDC+, we compared the results digitized from the speed improvement graph.

The speed improvement comparison results of the same MPI processor for EFDC-NIER and EFDC+ were similar, showing a speed improvement of a factor of four in four processors. When 16 processors were used, the improvement was a factor of 11.77 and 11 for EFDC-NIER and EFDC+, respectively, indicating that EFDC-NIER was slightly superior. When 32 processors were used, the improvement was a factor of 15 and 17 times for EFDC-NIER and EFDC+, respectively, demonstrating that EFDC+ was superior. In the case of EFDC+ MPI, the performance improvement effect increased as the number of processors

increased. However, the speed improvement varied according to the model configuration and hardware specifications. We determined that EFDC-NIER is better than or similar to EFDC+ MPI, especially considering that its improvement values include the simulation results of the water quality factors.

**Table A3.** Overview of EFDC-NIER and EFDC+ models and the hardware used.

| Category | EFDC-NIER | EFDC+ |
|---|---|---|
| | Nakdong River | Chesapeake Bay |
| Study area | 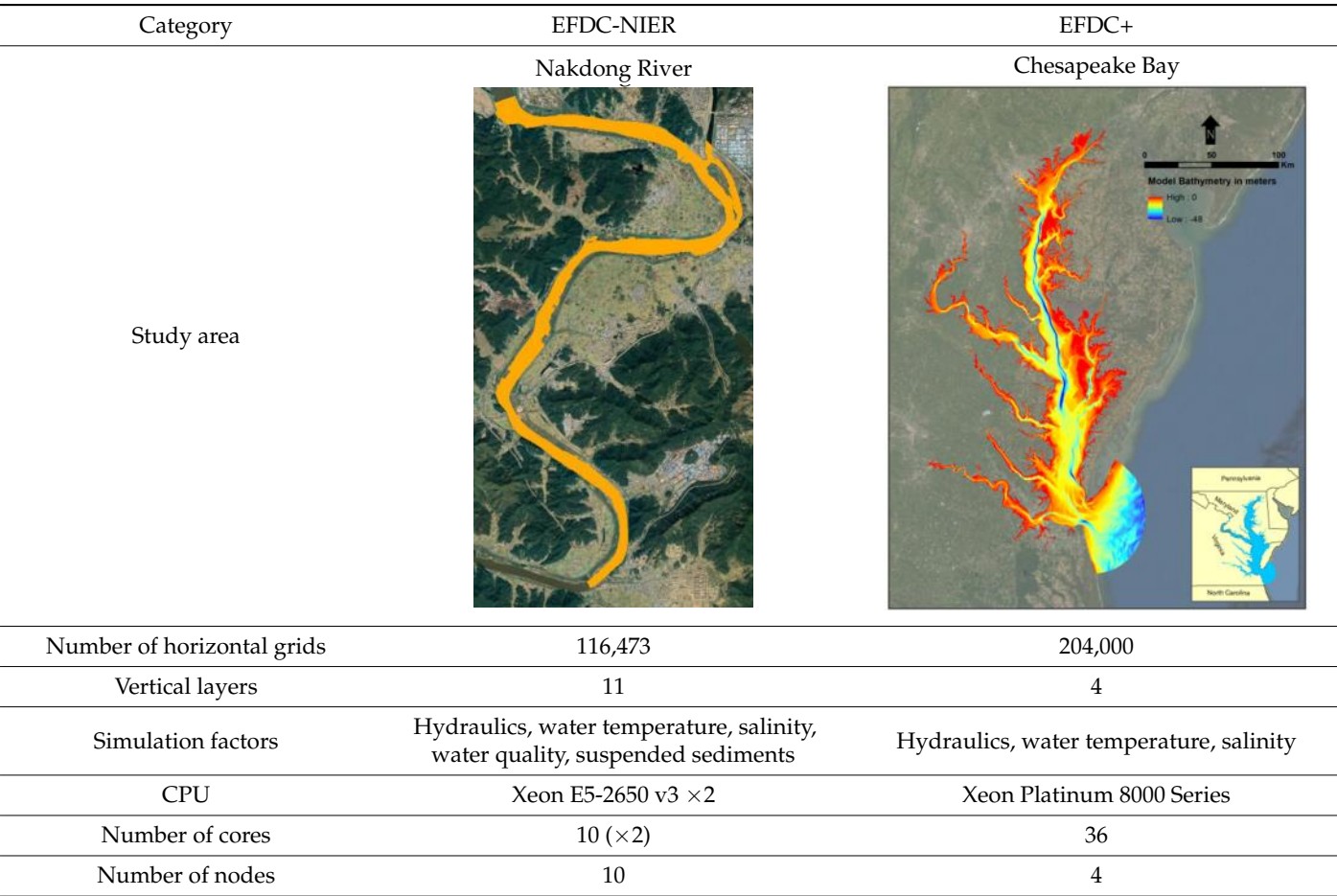 | |
| Number of horizontal grids | 116,473 | 204,000 |
| Vertical layers | 11 | 4 |
| Simulation factors | Hydraulics, water temperature, salinity, water quality, suspended sediments | Hydraulics, water temperature, salinity |
| CPU | Xeon E5-2650 v3 ×2 | Xeon Platinum 8000 Series |
| Number of cores | 10 (×2) | 36 |
| Number of nodes | 10 | 4 |

**Table A4.** Performance improvement in calculation speed by MPI processor.

| Category | MPI Processor (Unit for Speed Improvement: Times) | | | |
|---|---|---|---|---|
| | **4** | **8** | **16** | **32** |
| EFDC-NIER | 3.97 | 7.29 | 11.77 | 15.01 |
| EFDC+ | 4 | 7 | 11 | 17 |

**Appendix C**

For the performance evaluation of the parallel code, we performed OpenMP, MPI, and hybrid parallel performance evaluation for the model sets displayed in Table A3. The performance evaluation test was conducted using up to four nodes and 48 CPUs, considering the size of the model area. It was performed based on a 1-h computation, after which the calculation times were compared. Here, the combinations of the processors used in the OpenMP, MPI, and hybrid parallel performance evaluations were determined by considering the number of model grids and the hardware used in the experiment.

Table A5 shows the cluster for the parallel performance evaluation. The cluster consists of one login node and ten calculation nodes. Each calculation node is equipped with two units of the Intel Xeon CPU E5-2650 in the CentOS 6.7 operating system, and the data

communication is based on the Fourteen Data Rate (FDR) InfiniBand. Intel Parallel Studio 2017.1.043 was used for the creation and execution of the hybrid parallel program, and the combination of Intel OpenMP and Intel MPI 2017.1.132 was used for the OpenMP and MPI libraries, respectively. In the case of OpenMP, up to 20 processors can be used because 20 cores are configured at the maximum per node. However, the test was conducted using only up to 16 processors because the basic process of the operating system was running. In the case of MPI and the hybrid method, the processors were configured with a combination based on the number of nodes and the number of cores per node, and the test was conducted using 32 or 40 processors depending on the number of model grids.

Figures A1 and A2 show the ideal and actual values of the calculation times when the execution times of the model are compared. Here, the ideal value is the theoretical calculation time based on the increase in the number of calculation processors. For the ideal value, we applied the performance improvement–time ratio equation, which is known as Amdahl's law [36] and is expressed as

$$S = \frac{1}{(1-f) + f/n} \tag{A1}$$

where $S$ is the improvement ratio of the calculation speed, $f$ is the proportion of the total calculation time that the improved part occupies, and $n$ is the number of processors. For convenience of comparison, we assumed that $f$ was 1. Furthermore, we applied Equation (A1) by expressing Equation (A2) as the reduction ratio of the execution time:

$$TCT = \left( (1-f) + \frac{f}{n} \right) \times ACT = \frac{1}{n} \times ACT, \tag{A2}$$

where $TCT$ is the theoretical calculation time and $ACT$ is the actual calculation time.

**Table A5.** Specification of the parallel performance evaluation system.

| Category | | Cluster |
|---|---|---|
| CPU | Product | Intel Xeon CPU E5-2650 v3 * 2 ea |
| | #Cores | 10 (Total 20) |
| | Frequency | 2.30 GHz |
| | Cache | 25 MB |
| | Instruction | 64-bit |
| | Extension | Intel AVX2 |
| | Memory | 64 GB |
| | OS | CentOS release 6.7 |
| | Network | InfiniBand ConnectX-3 VPI FDR, IB (56 Gb/s) |
| | Compiler | Intel Parallel Studio 2017.1.043 |
| | OpenMP | Intel OpenMP |
| | MPI | Intel MPI 2017.1.132 |

(1) **OpenMP Parallel Performance Evaluation.**

In the OpenMP parallel computation evaluation, the performance was analyzed based on the number of OpenMP threads. Because OpenMP was scalable on a single node, eight experimental combinations of OpenMP threads were configured from 1 to 16. Table A6 and Figure A1 present the execution time for each function based on the number of OpenMP threads. As the number of threads increased, the OpenMP parallel calculation did not exhibit a calculation speed improvement close to the ideal value.

When examined based on the total time (HDMT2T), two threads showed a performance improvement of a factor of 1.7, and 12 threads showed an increase by a factor of 3.1. For most functions, the execution time decreased as the number of threads increased, but the increase was significantly lower.

**Table A6.** Execution time for each function based on the number of OpenMP threads.

| Category | Number of OpenMP Threads | | | | | | | |
|---|---|---|---|---|---|---|---|---|
| | 01 | 02 | 03 | 04 | 06 | 08 | 12 | 16 |
| HDMT2T | 656,953 | 385,124 | 308,917 | 259,871 | 231,408 | 221,675 | 215,195 | 210,575 |
| CALAVB | 21,823 | 11,475 | 8013 | 6445 | 5056 | 4282 | 3425 | 3083 |
| CALTSXY | 1722 | 1053 | 802 | 746 | 726 | 743 | 751 | 752 |
| CALEXP2T | 52,024 | 29,061 | 21,940 | 18,948 | 17,606 | 17,201 | 16,832 | 16,910 |
| CALCSER | 1006 | 1013 | 1148 | 1022 | 1059 | 1091 | 1088 | 1085 |
| CALPUV2C | 19,746 | 13,194 | 10,790 | 9814 | 9245 | 8976 | 8742 | 8842 |
| ADVANCE | 9040 | 7050 | 6393 | 6352 | 6367 | 6323 | 6266 | 6228 |
| CALUVW | 54,457 | 32,161 | 25,360 | 22,692 | 20,167 | 18,573 | 17,323 | 16,409 |
| CALCONC | 92,621 | 53,055 | 42,789 | 36,727 | 32,873 | 31,405 | 30,168 | 29,506 |
| SEDIMENT | 1931 | 1001 | 712 | 557 | 425 | 360 | 294 | 278 |
| WQ3D | 283,800 | 169,636 | 141,214 | 114,729 | 101,733 | 98,847 | 98,500 | 96,071 |
| CALBUOY | 6102 | 4293 | 3823 | 3716 | 3672 | 3619 | 3637 | 3691 |
| NLEVEL | 1781 | 971 | 678 | 549 | 486 | 489 | 491 | 505 |
| CALHDMF | 37,468 | 19,858 | 14,267 | 11,580 | 9188 | 8022 | 6687 | 6335 |
| CALTBXY | 10,391 | 5492 | 3768 | 2973 | 2278 | 1857 | 1446 | 1257 |
| QQSQR | 3063 | 1667 | 1199 | 997 | 786 | 673 | 559 | 555 |
| CALQQ2T | 55,889 | 29,635 | 21,593 | 17,561 | 15,242 | 14,577 | 14,262 | 14,231 |
| SURFPLT | 16 | 17 | 17 | 17 | 18 | 20 | 21 | 24 |
| VELPLTH | 38 | 37 | 38 | 37 | 39 | 43 | 48 | 55 |
| SALPTH | 726 | 824 | 827 | 861 | 864 | 878 | 862 | 883 |
| EEXPOUT | 3184 | 3499 | 3416 | 3423 | 3444 | 3448 | 3483 | 3536 |

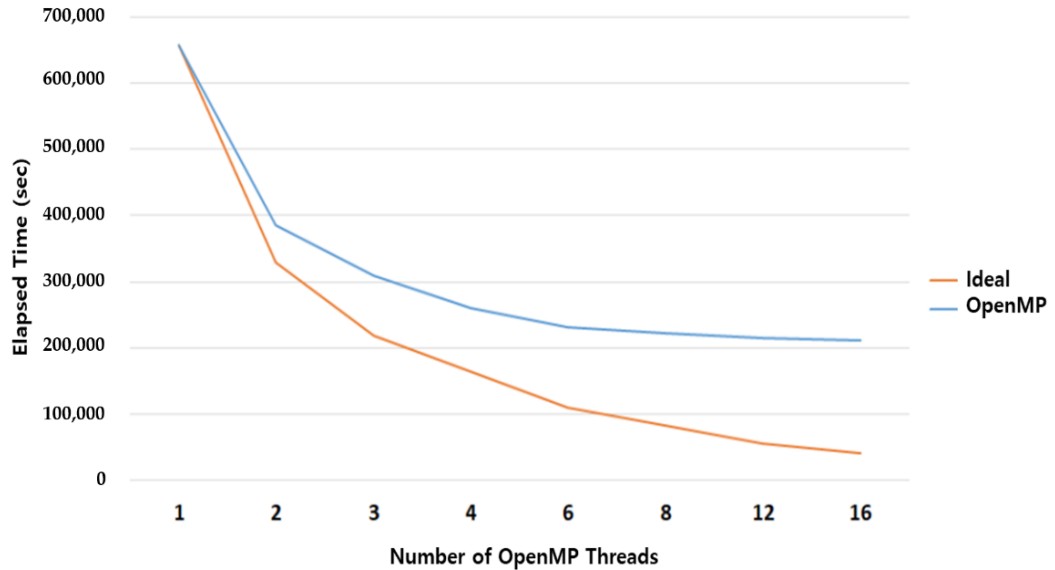

**Figure A1.** Execution time as a function of the number of OpenMP threads (detailed grid #2 of Nakdong River).

(2)　**MPI Parallel Performance Evaluation.**

In the MPI parallel computation evaluation, the performance was analyzed based on the number of MPI nodes. Because MPI was scalable on multiple nodes, four calculation nodes were used to configure seven experimental combinations for the number of MPI nodes from 1 to 40. Table A7 and Figure A2 present the execution times for each function based on the number of MPI nodes. The MPI parallel computation showed that the calculation speed improvement approached the ideal value as the number of nodes increased.

When examined based on the total time (HDMT2T), a performance improvement of a factor of 4.0 was exhibited with four MPI nodes, 11.8 with 16 MPI nodes, and 16.8 with 40 MPI nodes, reaching the maximum improvement. The MPI communication time (COMMUNICATION) did not increase as the number of MPI nodes increased; the elapsed time was 6605 s, which was the maximum when the number of MPI nodes was 12. In the MPI parallel performance, the parallel scalability of CALTSXY and CALPUV2C was not high. In contrast, CALAVB, CALUVW, CALCONC, and WQ3D, which have extensive calculations, showed larger performance improvements as the number of MPI nodes increased.

**Table A7.** Execution time for each function based on the number of MPI nodes (detailed grid #2 of Nakdong River).

| Category | Number of MPI Nodes | | | | | | |
|---|---|---|---|---|---|---|---|
| | 1 | 2 | 4 | 8 | 16 | 32 | 40 |
| HDMT2T | 656,953 | 323,191 | 165,614 | 90,132 | 55,814 | 43,759 | 40,212 |
| CALAVB | 21,823 | 10,987 | 5606 | 2816 | 1435 | 857 | 703 |
| CALTSXY | 1722 | 860 | 487 | 290 | 179 | 124 | 108 |
| CALEXP2T | 52,024 | 25,490 | 12,655 | 6546 | 3541 | 2365 | 2006 |
| CALCSER | 1006 | 1018 | 995 | 1015 | 1122 | 1996 | 1936 |
| CALPUV2C | 19,746 | 9959 | 5517 | 3125 | 2020 | 1628 | 1733 |
| ADVANCE | 9040 | 4205 | 2126 | 1107 | 656 | 592 | 456 |
| CALUVW | 54,457 | 26,806 | 13,885 | 6912 | 3739 | 2350 | 1879 |
| CALCONC | 92,621 | 46,994 | 23,973 | 13,098 | 7977 | 7168 | 6571 |
| SEDIMENT | 1931 | 954 | 477 | 239 | 124 | 72 | 57 |
| WQ3D | 283,800 | 130,926 | 63,214 | 30,843 | 15,659 | 9898 | 8316 |
| CALBUOY | 6102 | 3123 | 1623 | 765 | 404 | 321 | 252 |
| NLEVEL | 1781 | 293 | 146 | 76 | 41 | 30 | 23 |
| CALHDMF | 37,468 | 17,835 | 12,250 | 9587 | 10,347 | 9145 | 9478 |
| CALTBXY | 10,391 | 4420 | 2206 | 1105 | 577 | 332 | 274 |
| QQSQR | 3063 | 1444 | 732 | 375 | 197 | 122 | 99 |
| CALQQ2T | 55,889 | 27,152 | 13,673 | 7129 | 3762 | 2349 | 1920 |
| SURFPLT | 16 | 18 | 17 | 17 | 17 | 19 | 19 |
| VELPLTH | 38 | 39 | 37 | 38 | 37 | 42 | 46 |
| SALPTH | 726 | 376 | 183 | 88 | 55 | 55 | 40 |
| EEXPOUT | 3184 | 3641 | 3474 | 3438 | 3481 | 3980 | 3988 |
| COMMUNICATION | 0 | 6606 | 2291 | 1497 | 413 | 278 | 272 |

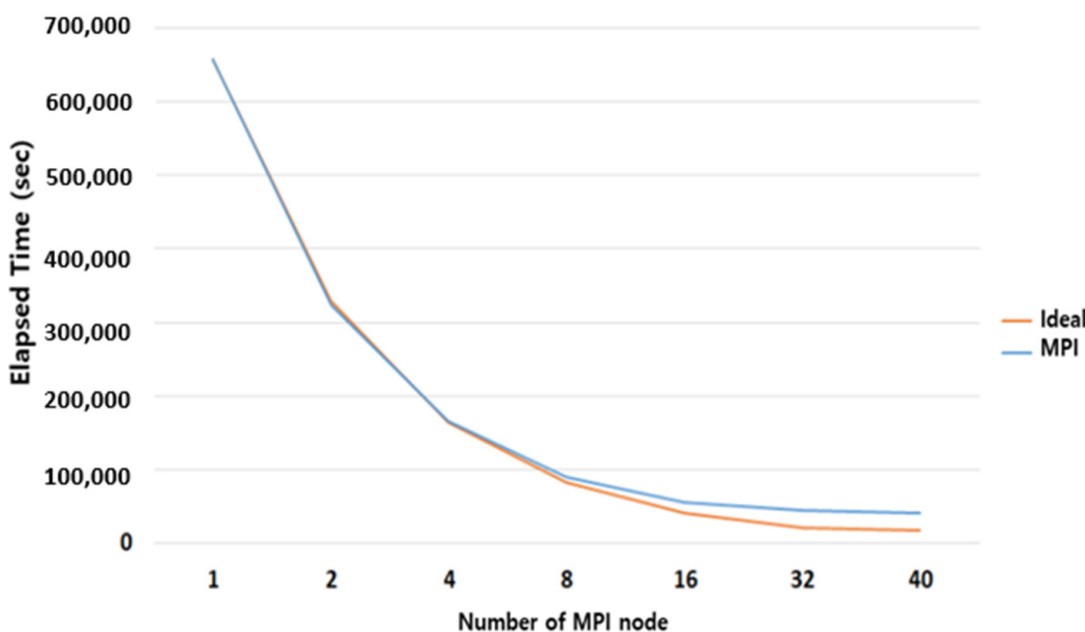

**Figure A2.** Execution time as a function of the number of MPI nodes.

(3) **Hybrid Parallel Performance Evaluation.**

In the evaluation of the hybrid parallel computation, the performance was analyzed based on the combinations of MPI and OpenMP threads. For quantitative evaluation, we composed four experimental hybrid combinations by fixing the number of utilizable CPUs to 40. In each case, the number of MPI nodes was increased from 4 to 8, 20, and 40. At the same time, the number of OpenMP threads was reduced from 10 to 5, 2, and 1. These particular numbers were chosen so that if the number of MPI nodes was multiplied by the number of OpenMP threads, the result would be 40.

As shown in Table A8, the best performance was achieved when the number of MPI nodes was eight and the number of OpenMP threads was five. When the computational time was examined for each function, no particular combination was superior. No consistency was observed in terms of time. For example, CALEXP2T and CALPUV2C showed the shortest execution time when the number of MPI nodes was eight and the number of OpenMP threads was five. However, in a combination of 40 MPI nodes and one OpenMP thread, the functions that require long computational times, such as CALUVW, SEDIMENT, and WQ3D, exhibited the best performance.

**Table A8.** Execution time for each function based on the hybrid combination.

| Category | Hybrid (MPI + OpenMP) Combination (40 cpu) | | | | | | | |
|---|---|---|---|---|---|---|---|---|
| | **MPI** | **OMP** | **MPI** | **OMP** | **MPI** | **OMP** | **MPI** | **OMP** |
| | **4** | **10** | **8** | **5** | **20** | **2** | **40** | **1** |
| HDMT2T | 54,457 | | 39,300 | | 41,800 | | 40,212 | |
| CALAVB | 975 | | 752 | | 765 | | 703 | |
| CALTSXY | 213 | | 122 | | 135 | | 108 | |
| CALEXP2T | 3411 | | 1921 | | 2126 | | 2006 | |
| CALCSER | 1078 | | 1079 | | 1307 | | 1936 | |
| CALPUV2C | 3224 | | 1724 | | 2128 | | 1733 | |
| ADVANCE | 1350 | | 515 | | 436 | | 456 | |
| CALUVW | 3293 | | 2283 | | 2213 | | 1879 | |
| CALCONC | 8102 | | 5851 | | 5816 | | 6571 | |

**Table A8.** *Cont.*

| Category | Hybrid (MPI + OpenMP) Combination (40 cpu) | | | | | | | |
|---|---|---|---|---|---|---|---|---|
| | MPI | OMP | MPI | OMP | MPI | OMP | MPI | OMP |
| | **4** | **10** | **8** | **5** | **20** | **2** | **40** | **1** |
| SEDIMENT | 155 | | 88 | | 73 | | 57 | |
| WQ3D | 17,240 | | 11,109 | | 10,090 | | 8316 | |
| CALBUOY | 1016 | | 357 | | 245 | | 252 | |
| NLEVEL | 122 | | 42 | | 27 | | 23 | |
| CALHDMF | 4039 | | 5794 | | 8882 | | 9478 | |
| CALTBXY | 324 | | 279 | | 263 | | 274 | |
| QQSQR | 152 | | 104 | | 100 | | 98 | |
| CALQQ2T | 3738 | | 2065 | | 2295 | | 1920 | |
| SURFPLT | 20 | | 19 | | 19 | | 19 | |
| VELPLTH | 43 | | 45 | | 43 | | 46 | |
| SALPTH | 178 | | 57 | | 32 | | 40 | |
| EEXPOUT | 3625 | | 3584 | | 3516 | | 3988 | |
| COMMUNICATION | 2101 | | 1477 | | 1256 | | 272 | |

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
