# Peer review of "Parallelization of a 3-Dimensional Hydrodynamics Model Using a Hybrid Method with MPI and OpenMP"

_processes, doi:10.3390/pr9091548_

Round 1
Reviewer 1 Report
In the proposed manuscript, the authors detail the optimization of parallelization in a CFD code focusing on water pollution evaluation. According to the results provided, performance gains were achieved, leading to faster decision-making in water quality control.
The subject under consideration is of interest and scientifically relevant. However, I believe there is a lack of detail in the discussions and a lack of references to support the authors' statements. The manuscript quality could be improved if the authors could compare the computation time optimization of the proposed code concerning other software also in use. In this way, the reader could easily infer the advantages.
Further comments:
- The initial part of the introduction lacks references to support the arguments made by the authors. The citation to the literature starting on line 65 should be rewritten since references are presented in a bullet point-like fashion.
- In line 75, the authors define parallel calculation as an efficient calculation technique. What do the authors mean by this? Is 'efficient' related to a compromise between processing speed and result quality, or is it meant to indicate accurate results?
- Between lines 97-110, the authors introduce to the discussion several computational tools without providing any reference related to their development. This should be addressed.
- While Figure 1 makes an interesting comparison, the authors need to address the abbreviations used explicitly.
Reviewer 2 Report
The manuscript highlights an interesting application of parallel hydraulical simulation codes. In principle it fits well into the journal "processes". Before it could be published, a significant number of issues should be improved including:
a) It is mentioned that an existing code EFDC-NIER is applied in the current study. Is this the open source code version refered to in reference [23]? If yes, please refer clearly to the repository inlcuding the applied version where it could be downloaded (already on page -3-, line 114). Please provide most important details concerning the code (models, software concept, programming language etc.) as it could not be assumed that the code is known to all the readers.
aa) Are there any additional studies concerning efficiency (strong/weak scaling tests) of the applied code. Please refer to them!
b) Figure 1: Most abbreviations in the figure are not explained. Thus, the figure could not be understood in detail, please explain abbreviations.
c) Figure 2 and the related explanation in the text should be improved significantly: What is meant with "fixed partition"? The red and blue rsults refer to the applied solver as I guess. Please clarify. If scaling isshown, you should refer to technical details (kind of CPUs, how many core/CPU, How many CPU/node, how are the nods connected etc.).
dd) Figure 2: I could not follow your explanation in the text. When the parallel version is applied, The parallel efficiency drops to around 40% if 25 processors are used. In the text you mentioned an efficienfy of 70% - please clarify!
e) Please explain (results section -3-), why you did not perform simulations on HPC environments. I understand, that you are aiming for a comparison for rather small compute clusters (MPI vs. OpenMP). Anyway, it would be important to understand how the code scales for a much larger number of nodes.
f) Please explain shortly (in the appendix) what kind of partial differential equation are solved. Please also mention the numerical method (finite volumes/differences/elements etc.)
Round 2
Reviewer 2 Report
I appreciate the work of the authors in improving the quality of the manuscript. Unfortunately, still some issues are opne and should be implemented before the paper could be published:
I refer here to my nomenclature (numbering of the 1st round of review)
a) I asked to give more information about models, software concept, programming language etc. Unfortunately, the authors did not include any of that information yet. Please add the mentioned details. Please also refer to the original repository of EDFC. For me it's not clear (I do not know the code license) if the modified version of EDFC-NIER is allowed to be distributed e.g. on a repository of the authors. Please clarify this important issue. If yes, the modified source code should be provided to the community.
aa) Please mention explicitely in the paper that strong and weak scaling tests are not available yet and that this is an open task for the future. Why are these tests not provided yet? Concerning efficiency, such investigations are fundamental. Please explain!
f) In my question, I was asking about the underlying mathematical model of EDFC and it's numerical treatment with finite volumes. Please explain that important issue. I asked about that information already in the 1st review round.
